# SIRI: Spatial Relation Induced Network For Spatial Description Resolution

**Peiyao Wang** †     **Weixin Luo** †     **Yanyu Xu**
ShanghaiTech University
{wangpy, luowx, xuyy2}@shanghaitech.edu.cn

**Haojie Li**
Dalian University of Technology
hjli@dlut.edu.cn

**Shugong Xu**
Shanghai University
shugong@shu.edu.cn

**Jianyu Yang**
Soochow Univerisity
jyyang@suda.edu.cn

**Shenghua Gao**
gaoshh@shanghaitech.edu.cn

## Abstract

Spatial Description Resolution, as a language-guided localization task, is proposed for target location in a panoramic street view, given corresponding language descriptions. Explicitly characterizing an object-level relationship while distilling spatial relationships are currently absent but crucial to this task. Mimicking humans, who sequentially traverse spatial relationship words and objects with a first-person view to locate their target, we propose a novel spatial relationship induced (SIRI) network. Specifically, visual features are firstly correlated at an implicit object-level in a projected latent space; then they are distilled by each spatial relationship word, resulting in each differently activated feature representing each spatial relationship. Further, we introduce global position priors to fix the absence of positional information, which may result in global positional reasoning ambiguities. Both the linguistic and visual features are concatenated to finalize the target localization. Experimental results on the Touchdown show that our method is around 24% better than the state-of-the-art method in terms of accuracy, measured by an 80-pixel radius. Our method also generalizes well on our proposed extended dataset collected using the same settings as Touchdown. The code for this project is publicly available at https://github.com/wong-puiyiu/siri-sdr.[1]

## 1  Introduction

Visual localization tasks aim to locate target positions according to language descriptions, where many downstream applications have been developed such as visual question answering (1; 22; 23), visual grounding (20; 18; 28; 6) and spatial description resolution (SDR) (3), etc. These language-guided location tasks can be categorized in terms of input formats, e.g. perspective images in visual grounding or panoramic images in the recently introduced SDR.

**The Challenge of SDR:** Both of visual grounding and spatial description resolution tasks need to explore the correlation between vision and language to locate the target locations. Unlike traditional visual grounding, the recently proposed spatial description resolution of panoramic images, however, presents its own difficulties due to the following aspects. (1) As shown in Figure 1, the complicated

entities, such as buildings in an image, present some challenges for the advanced object detection (9). For example, existing methods may fail to instantiate multiple adjacent buildings. (2) The short language descriptions in visual grounding are more about well-described instances with multiple attributes, while the long descriptions in spatial description resolution describe multiple spatial relationship words, such as 'your right/left', 'on the left/right' and 'in the front', from a distant starting point to the target. It is worth noting that such crucial issues have not been well addressed in previous work. (3) Panoramic images in visual grounding with a first-person view cover more complex visual details on a street compared to the perspective images with a third-person view in visual grounding.

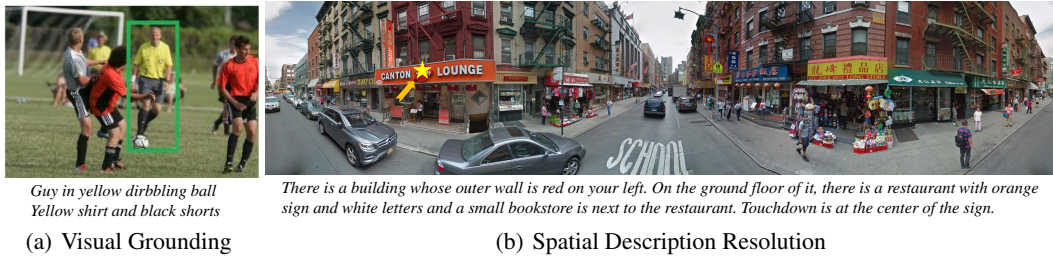

*Guy in yellow dirbbling ball*
*Yellow shirt and black shorts*

(a) Visual Grounding

*There is a building whose outer wall is red on your left. On the ground floor of it, there is a restaurant with orange sign and white letters and a small bookstore is next to the restaurant. Touchdown is at the center of the sign.*

(b) Spatial Description Resolution

Figure 1: Examples of the datasets for the recently proposed spatial description resolution and conventional visual grounding. For the visual grounding illustrated on the left, there are simple entities in the image with a third-person view and the language descriptions are also simple and short. Regarding SDR on the right, real-world environments with a first-person view contain comprehensive entities. The corresponding languages have multiple entities and spatial relationships. The yellow star in the panorama on the right illustrates the target location according to the language descriptions.

**Our Solution:** To efficiently tackle SDR, humans start at their own position with a first-person perspective and sequentially traverse the objects with spatial relationship words, finally locating their target. To mimic the human behavior on SDR, we propose a spatial relationship induced (SIRI) network to explicitly tackle the SDR task in a real-world environment. As shown in Figure 2, we firstly leverage a graph-based global reasoning network (5) (GloRe) to model the correlations of all the object-object pairs in the extracted visual feature, where the visual feature is projected to a latent space to implicitly represent object instances in an unsupervised manner. Implicitly learning object concepts and their visual correlations free us from explicitly designing an object detector for a street view. Meanwhile it enables each object in the image to accumulate its contextual information, which is extremely important for scene understanding as well as for spatial description resolution. Next, a local spatial relationship guided distillation module is appended to distill the visual features to different discriminative features, where each corresponds to a spatial relationship word. We argue that distilling visual features with local spatial relationships concentrates on specific features corresponding to these crucial language hints, consequently facilitating final target localization. After averaging all the distilled features, we introduce two global coordinate maps, of which the origin is at the agent's position, i,e., the bottom center of the image. Such a position prior alleviates the ambiguities of global positional reasoning in an efficient way. All encoded linguistic features, distilled visual features and position priors are fed into LingUnet(3) to finalize target localization. It is worth noting that our solution tackles the task of SDR in a highly efficient way and performs significantly better than other existing methods.

**Our contributions:** (1) A novel framework is proposed to explicitly tackle the SDR task in terms of object-level visual correlation, local spatial relationship distillation and global spatial positional embedding. (2) Extensive experiments on the Touchdown dataset show that our method outperforms LingUnet (3) by 24% in terms of accuracy, measured by an 80-pixel radius. (3) We propose an extended dataset collected using the same settings as Touchdown, and our proposed method also generalizes well.

## 2 Related Work

**Language Guided Localization Task.** Visual grounding (20; 18; 28; 6) and referring expression comprehension (16; 27; 25) aim to locate target objects or regions according to given languages. The images in these tasks are perspective images that contain a limited number of entities, and the expression languages are also short. Object detection, which is one of the tasks in these datasets, is commonly used to provide a prior that functions as a correspondence between objects in images and language-based entity nouns. Methods under the object detection framework can be categorized in two ways. The first category (19; 24; 27; 18) has two stages, in which object detection is carried out at the beginning and object proposals are ranked according to the language query. Two-stage approaches, however, are time-consuming. Thus, one-stage approaches (26; 21; 29; 4) have been proposed to achieve greater efficiency. Nevertheless, the object detectors can fail when it comes to real-world environments in spatial resolution description (3), where more objects and complex backgrounds are included with large fields of view, as shown in Figure 1. In addition to this, the given language descriptions in SDR are longer and describe more object pair spatial relationships. Undoubtedly, existing one-stage methods with weak contextual information on objects for grounding do not specialize when processing spatial positioning words. Recently, LingUnet (3) was proposed, and it treats linguistic features as dynamic filters to convolve visual features, taking all regions into consideration. But it does not yet fully explore the visual and spatial relationships in such complex environments. In this paper, we intend to fully investigate these spatial relationships between objects.

**Spatial Positional Embedding.** As has been studied, convolutional layers cannot easily extract position information (14). Thus, spatial positional embedding has been commonly used in localization tasks. For instance, Liu (14) proposed CoordConv to concatenate coordinate maps into channels of features, enabling convolutions to access their own input coordinates, which was of ultimate benefit to multiple downstream tasks. In addition, coordinate maps have been embedded in object detection (8). An 8-D spatial coordinate feature is provided at each spatial position for image segmentation (7). (15) included 2D coordinates maps with the corresponding regions to predict more precise depth maps. (2) concatenated positional channels to an activation map to introduce explicit spatial information for recognition tasks. In SDR, it is particularly important to accurately describe the positional information of each object since the corresponding language descriptions sequentially depict bearings between objects. All these operations in the recently proposed LingUnet are, however, convolutional layers, leading to an absence of positional information for each pixel. Undoubtedly, ambiguities emerge when duplicated target objects are present in the same image. Unfortunately, spatial positional embedding has not been properly studied in SDR. Thus, we introduce a global spatial positional information to SDR to handle this problem.

## 3 Method

### 3.1 Overview

We illustrate our proposed SIRI network in Figure 2. It consists of a visual correlation, a local spatial relationship guided distillation and a global spatial positional embedding. For privacy preserving, only the features extracted from a pretrained RESNET18 (10) are provided in the TouchDown dataset. Thus, given the object-level visual feature of an image $I$ with a shape of $h$(height) $\times w$(width) and a natural language description, the output of our proposed SIRI network is a heatmap with the same resolution of the input image, and its peak is the final target localization result. All spatial relationships represented by orientation words in the entire dataset form the set $W = \{$right, left, $...\}$. Further, we denote the orientation words in the descriptions corresponding to the image as $W_I$.

**Language representation.** Different images have different language description lengths. We adapt a Bidirectional LSTM (BiLSTM) to extract the linguistic features at all time steps for all words in a given language description. Then an averaging operation is conducted on these linguistic features, resulting in a fixed-length feature vector $L_I$. It then functions as a dynamic filter in LingUnet, where it will be separated into two equally sized slices, after which each slice will be projected and reshaped into a filter by a fully-connected layer. Also, it will be projected and reshaped into the linguistic features used in feature concatenation via another full-connected layer.

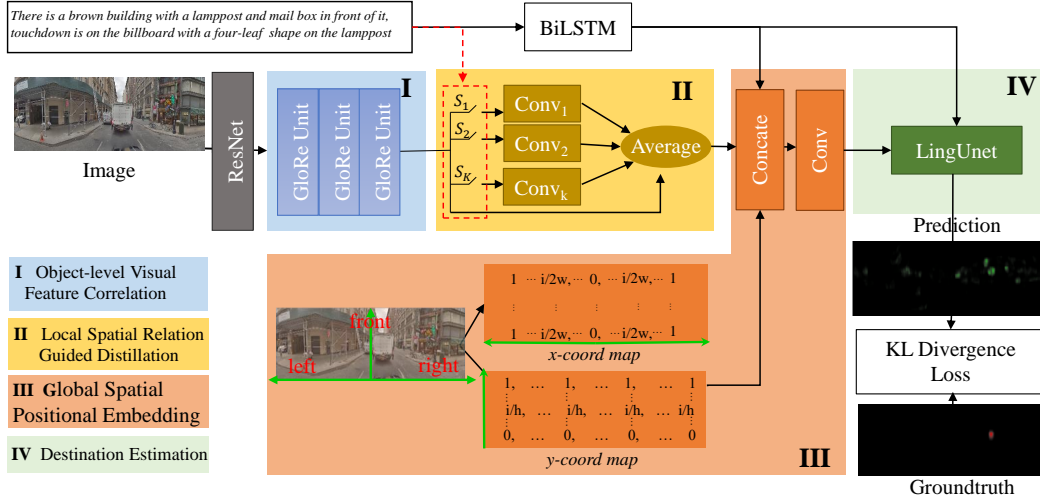

Figure 2: The entire framework of our proposed SIRI for SDR. The orientation words in the descriptions function as switches, i.e. if a certain orientation word is included in the descriptions, then the corresponding $Conv_k$ branch in Stage II is activated once regardless of the number of appearances. Otherwise, the corresponding branch is hibernated. This figure is best viewed in color.

## 3.2  Object-level Visual Feature Correlation

Since the goal of SDR is to localize a specific position on an object, an advanced object detector such as Mask RCNN (9) can be used to instantiate each object in the given street-view image. Cohesive buildings cannot be differentiated by such detectors, however, leading to a single detection box on them. Therefore we instantiate each object in a latent space using GloRe (5). Specifically, we cast the input space into an interaction space by multiplying a projection matrix, where a graph convolutional network (13) is then utilized to conduct visual relationship learning. Another projection matrix is then used to cast the visual relationship features back to the original space. Such an embedding space enables us to conduct an object-level visual feature correlation in an unsupervised manner. Thus, we stack the GloRe multiple times to fully explore the visual relationships of the input visual features.

## 3.3  Local Spatial Relation Guided Distillation

The correlated visual features $X$ are overly dense, and they contain a maximum of visual features of objects and local spatial relationships. This makes it more difficult to find the target position. On the other hand, the spatial relationship words in the corresponding language descriptions are limited and are related to some specific objects. Thus, distilling the specific spatial relationships corresponding to these language descriptions is helpful in locating the language-guided regions in panoramas. In this paper, we employ spatial relationship guided distillation to distill the correlated visual features based on each spatial relationship (orientation) word in the language descriptions. Specifically, we introduce $K$ branches of convolutional blocks that correspond to $K$ orientation words. For each branch corresponding to a specific orientation word, a different $5 \times 5$ trainable filter is used to activate the features corresponding to a specific word. Ideally, $K$ would be equal to the size of $W$. This is, however, impractical for handling the huge sets of orientation words in the entire dataset. Thus, we select the top $k$ high-frequency words among $W$, which forms the set $W^H$. Then, the output of these $k$ branches will be averaged. We also use a skip-connection to add the input features $X$ to this output in case none of the high-frequency orientation words are present in the language descriptions. Mathematically, the output of the spatial relationship distillation $G$ can be formulated as follows:

$$G(X) = \sum_{k=1}^{K} \mathbb{1}_{\{W_I^k \in W^H\}} \times \text{Conv}_k(X) + X \tag{1}$$

.

As shown in Figure 2, The orientation words in the descriptions function as switches, which means if a certain orientation word is present in the descriptions, then the corresponding convolutional branch is activated regardless of the number of appearances. Finally, the features corresponding to the high-frequency orientation words in the descriptions will be distilled in an end-to-end training manner.

### 3.4 Global Spatial Positional Embedding

The previous procedure, however, misses the global positional information, which makes the final target localization difficult due to the global positional words such as 'on your left' in the language descriptions. To introduce this absent but extremely important information, we introduce a spatial positional embedding with global coordinate maps to fix this. By concatenating the distilled features with two auxiliary features, the ambiguities due to the absence of global positional information are alleviated.

**Global Coordinate Maps.** Since the language descriptions are based on the egocentric viewpoint of an agent that is always located at the bottom center and that is moving forward, most reasoning routes start from the position of the agent and turns to either the upper left side or the upper right side of the panorama. Thus, such orientations provide a strong orientation prior for reasoning routes. More specifically, we build a coordinate whose origin is the same with the location of the agent (the bottom center of the image), where the x-axis runs along the horizontal direction and the y-axis is the vertical direction. We then arrive at two coordinate maps whose values correspond to the coordinates in the x-axis direction and the y-axis direction, respectively, as shown in Figure 2. These coordinate maps are denoted as $M^C \in \mathbb{R}^{h^F \times w^F \times 2}$. In addition, we normalize them onto the range [0, 1].

Considering the following fusion of different feature maps, we firstly transform the language representation $L_I$ to a vector with a fully-connected ($FC$) layer and we then reshape it to a feature map with the same resolution of the image. Then, we concatenate these linguistic features with coordinate maps, as well as with the distilled visual features. This is followed by a convolution operation with a kernel size of $3 \times 3$ to fuse all of this information. Formally, we denote the output $R$ of these operations as follows:

$$R(M^C, L_I, I^F) = \text{Conv}([M^C; \text{Reshape}(FC(L_I)); G(I^F)]). \tag{2}$$

### 3.5 Destination Estimation

As this point, we can directly predict the target position map, which is regularized by the corresponding ground-truth. Motivated by the success of LingUnet for SDR, we append a LingUnet for destination estimation. Formally, we denote the predicted heatmap $\hat{M}$ as follows:

$$\hat{M} = \text{LingUnet}(R(M^C, L_I, I^F), L_I). \tag{3}$$

It is worth noting that our proposed method can achieve significantly better results compared to LingUnet, even without appending LingUnet.

### 3.6 Objective Function

Given the input image feature $I$ and the corresponding language description, we apply the Equation 3 to generate the predicted heatmap $\hat{M}$. For the ground-truth heatmap, we apply a gaussian filter over the target position and denote it as $M$. We then leverage a KL divergence loss between the ground-truth heatmap with an $h \times w$ down-sampling, after which the predicted heatmap $\hat{M}$ over each pixel is as follows:

$$L_{KL}(\hat{M}, M) = \sum_{i=1}^{hw} M_i \log \hat{M}_i. \tag{4}$$

## 4   Experiments

**Touchdown and Extended Touchdown datasets.** We conducted all experiments on the TouchDown dataset (3), which is designed for navigation and spatial description reasoning in a real-life environment. In this paper, we focus on the analysis of the spatial description resolution task of locating on

| Method | A@40px(%) ↑ | A@80px(%) ↑ | A@120px(%)↑ | Dist↓ |
|---|---|---|---|---|
| Validation Set / Testing Set | | | | |
| Random (3) | 0.18 / 0.21 | 0.59 / 0.78 | 1.28 / 1.89 | 1185 / 1179 |
| Center (3) | 0.55 / 0.31 | 1.62 / 1.61 | 3.26 / 3.39 | 777 / 759 |
| Average (3) | 1.88 / 2.43 | 4.22 / 5.21 | 7.14 / 7,96 | 762 / 744 |
| Text2Conv (3) | 24.03 / 24.82 | 29.36 / 30.40 | 32.60 / 34.13 | 195 / 182 |
| LingUnet (3) | 24.81 / 26.11 | 32.83 / 34.59 | 36.44 / 37.81 | 178 / 166 |
| SIRI-Conv | 43.47 / 44.51 | 53.62 / 55.73 | 64.16 / 65.26 | 114 / 107 |
| **SIRI** | **44.86 / 46.93** | **55.83 / 58.33** | **65.69 / 67.66** | **105 / 100** |

Table 1: Comparison with different methods on TouchDown's validation set and testing set.

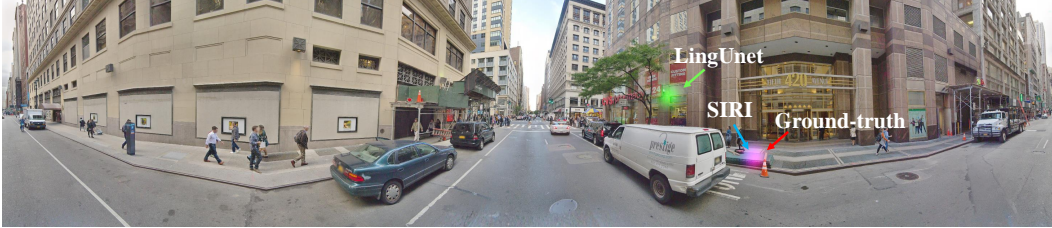

*In the street outside the building with the gold rotating doors is an orange traffic cone. Touchdown is sitting on top of the orange cone.*

Figure 3: Visualization of the predictions of SIRI and LingUnet, as well as the ground-truth. SIRI's predictions are closer than LingUnet's predictions to the ground-truth. This figure is best viewed in color.

Touchdown given panoramic images and corresponding language descriptions. Touchdown location strings of text are given as natural languages, and the locations are presented as heatmaps. In total, this dataset contains $27,575$ samples for SDR, including $17,878$ training samples, $3,836$ validation samples and $3,859$ testing samples. To see how well our proposed method generalizes in the wild, we built a new, extended dataset of Touchdown, using data collected under the same settings as the original Touchdown. The details and an analysis of our proposed dataset can be found in the supplementary materials.

**Implementation Details.** It should be noted that we do not conduct any down-sampling operations in any of the modules, which means that the resolutions of all the feature maps are $100 \times 464$. In the spatial relationship guided distillation procedure, we choose the top six of the high-frequency orientation words, because of the large number of orientation words in the entire dataset and their long-tail distribution. In all experiments, we use the Adam optimizer(12) to train the network. In addition, the number of training mini-batches and the learning rate are 10 and 0.0001 respectively. The code is implemented in Pytorch.

**Evaluation Metric**. Following the previous work (3), we adapt the same evaluation metric in terms of accuracy and distance. We denote the peaks of the predicted heatmap and the ground-truth heatmap as $\hat{m}$ and $m$, respectively. Formally, the distance is defined as follows:

$$Dist(m,\hat{m}) = \|m - \hat{m}\|_2 \tag{5}$$

Similarly, the accuracy is defined over the whole dataset with $N$ samples, based on radii $r$ of 40, 80 and 120 pixels, denoted as A@40px, A@80px and A@120px, respectively.

$$\text{A@}r\text{px} = \frac{1}{N}\sum_i^N \mathbb{1}_{\{Dist(m_i,\hat{m}_i)\leq r\}} \times 100\% \tag{6}$$

## 4.1 Comparison with the State-Of-The-Art

Following the previous work (3), we compare our method with three non-learning-based methods, i.e. Random, Center and Average, as well as two learning-based baselines, i.e. Text2Conv and LingUnet,

| Method | A@40px(%) ↑ | A@80px(%)↑ | A@120px(%)↑ | Dist↓ |
|---|---|---|---|---|
| LingUnet (3) | 22.41 | 30.72 | 34.57 | 178 |
| **SIRI** | **42.14** | **48.96** | **56.39** | **122** |

Table 2: Generalization results on our proposed extended Touchdown dataset.

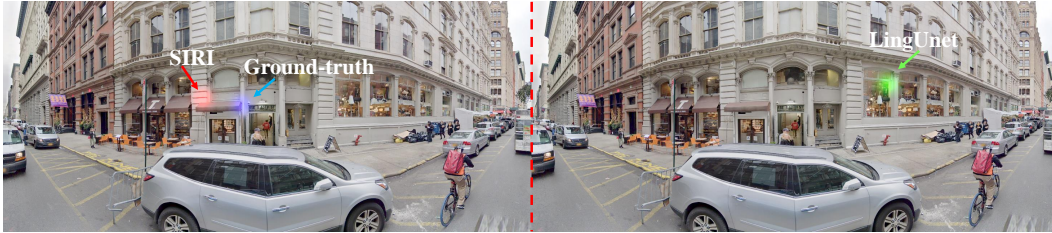

*Touchdown is centered on the front of the 4th brown awning on the left that is slightly wider than the other 3.*

(a) SIRI successfully predicts the target position while LingUnet fails in this case, even though the orientation word 'on the left' is provided.

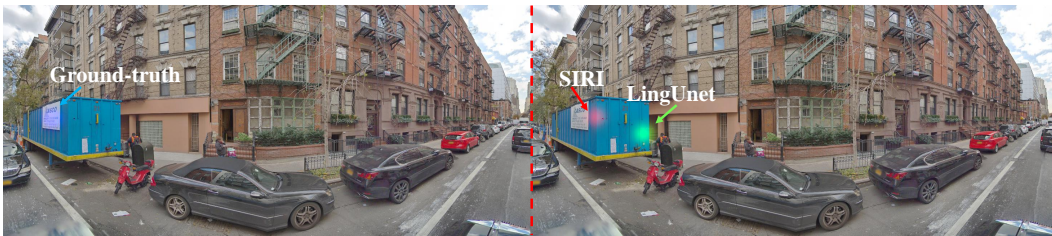

*A large blue dumpster-type structure. Move past it and turn back to face it. On the side is a large white sign. Touchdown is on this sign.*

(b) Both SIRI and LingUnet wrongly predict the target position since the orientation words such as 'on the side' are ambiguous.

Figure 4: A visualization of the predictions of SIRI and LingUnet, as well as the ground-truth when a copy-paste operation is conducted to introduce visual ambiguities. This figure is best viewed in color.

to prove the effectiveness of our proposed SIRI network. To investigate the effects of the introduced LingUnet on destination estimation, we replace LingUnet with several convolutional layers with nearly the same number of parameters as a SIRI-Conv baseline.

We show the results of all the methods on the Touchdown dataset in Table 1, where we can see that our proposed SIRI network significantly improves the performance by almost 24% for A@80px, with a decrease of 66 pixels of distance error decreasing in the testing set compared to the state-of-the-art method LingUnet. In addition, when the baseline SIRI-Conv replaces LingUnet with several convolutional layers, a slight performance drop results, demonstrating the effectiveness of our proposed modules in SIRI.

We also compare our approach to some closely related work, including FiLM (17). FiLM has achieved state-of-the-art performance on the CLEVR benchmark (11). The GRU module in FiLM functions similarly to the BiLSTM module in our proposed SIRI, and the FiLM functions similarly to the spatial relationship guided distillation module. In addition, positional embedding is leveraged in both studies. The spatial relationship guided distillation that we introduced, however, induces the network with a gate mechanism, while FiLM performs a feature-wise affine transformation. To fairly compare our SIRI with FiLM, we adopt a local spatial relationship guided distillation module and a global spatial positional embedding module only in SIRI. Our SIRI has an accuracy@80px of 58.33%, much better than FiLM, which has an accuracy@80px of 52.37%. This demonstrates the effectiveness of our SIRI.

## 4.2 Generalization of Our Proposed Dataset

As shown in Table 2, we also test our proposed model (trained on the entire TouchDown dataset) on our proposed extended dataset. This table shows that SIRI consistently outperforms LingUnet by around 18% for A@80px, which demonstrates the robustness of our proposed method. Further, we visualize the prediction results on our proposed extended Touchdown dataset, as shown in Figure 3, which illustrates how these predictions are closer to the ground-truth compared to those of LingUnet.

## 4.3 Ablation Study

Our proposed SIRI significantly improves the accuracy of reasoning target positions via the addition of our proposed modules, as compared to the baseline LingUnet (a). Knowing this, we carefully investigate the impact of each proposed module, as well as their combinations of them. To begin with, the modules, except LingUnet are evaluated independently, corresponding to the methods (b), (c) and (d). As shown in Table 3, the most improvement is from the local spatial relationship guided distillation module, where A@80px is improved by around 10%. And the second biggest improvement of 4% is from the local spatial positional embedding. This means that this module provides accurate positional information when exploring the spatial position relationship of objects. In addition to this, we carefully study the number of top-$k$ selected orientation words in this module. The accuracies for @80px are 51.02%, 58.33% and 59.74% when $k$ is 4, 6 and 8 respectively, and where the method (g) is evaluated. To reduce the time cost, we set $k$ to 6 throughout all the experiments.

Further, performance consistently increases when more modules are connected in series. This means that these modules are complementary and ultimately achieve state-of-the-art accuracy is achieved when they are all appended.

| Method | Procedure | | | Accuracy | | |
|--------|-----|-----|-----|------------|------------|-------------|
| | I | II | III | A@40px(%)↑ | A@80px(%)↑ | A@120px(%)↑ |
| (a) | | | | 26.11 | 34.59 | 37.81 |
| (b) | ✓ | | | 27.96 | 37.42 | 42.13 |
| (c) | | ✓ | | 31.05 | 44.95 | 54.83 |
| (d) | | | ✓ | 28.17 | 38.76 | 44.27 |
| (e) | ✓ | ✓ | | 33.71 | 45.63 | 56.21 |
| (f) | | ✓ | ✓ | 43.67 | 56.24 | 65.82 |
| (g) | ✓ | ✓ | ✓ | **46.93** | **58.33** | **67.66** |

Table 3: The ablation study for all procedures: (I) Object-Level Visual Feature Correlation; (II) Local Spatial Relationship Guided Distillation; (III) Global Spatial Positional Embedding. These procedures are sequentially appended to the LingUnet-only baseline.

## 4.4 Visual Ambiguities

We further carefully investigate which part of SDR our proposed SIRI improves. To begin , the SDR task can be split into two individual sub-tasks, i.e. target object localization and spatial relationship reasoning. It is worth noting that the performance gain in the SDR task may result merely from the improvement of target object localization, especially when the target object is unique throughout the entire given image and the spatial relationship reasoning is ignored. It is, however, difficult to visualize the spatial relationships. Thus, we copy the left half part of each testing image and paste it to the right half part of the image, when the target is on the left, and vice versa. Finally, we introduce visual ambiguities in all the testing images to see whether our proposed SIRI as well as LingUnet is able to capture the spatial relationships rather than only conduct object localization.

**Quantitive Analysis.** To this end, we firstly conduct inference for all the copy-pasted testing images, and we calculate the accuracies. As shown in Table 4, we observe a roughly 5% drop for A@40px over SIRI, compared to around a 14% drop over LingUnet. We claim that our proposed SIRI fully explores these spatial relationships, thus leading to a smaller performance drop when visual ambiguities are introduced.

| Method | A@40px(%) ↑ | A@80px(%)↑ | A@120px(%)↑ |
|---|---|---|---|
| original / copy-paste / performance drop | | | |
| LingUnet (3) | 26.11 / 12.08 / **14.03** | 34.59 / 27.62 / **6.97** | 37.81 / 32.46 / **5.35** |
| SIRI | 46.93 /42.19 / **4.74** | 58.33 / 52.86 / **5.47** | 67.66 / 60.25 / **7.41** |

Table 4: Performance drop caused by the introduced visual ambiguities for SIRI and LingUnet.

| Method | #Para($\times 10^7$) | Time(s) | A@80px(%) ↑ |
|---|---|---|---|
| LingUnet | 1.428 | 8.62 | 34.59 |
| **SIRI-SDR** | 1.819 | 21.35 | 58.33 |

Table 5: The running time, number of parameters and the A@80px for SIRI and LingUnet on the Touchdown dataset.

**Qualitative Analysis.** Next, we visualize the localization results of SIRI and LingUnet in two cases as follows, to see what caused this performance drop.

1) For those samples whose language descriptions have some absolute orientation words such as 'on your left', this introduced visual ambiguity does not adversely affect target localization. As shown in Figure 4 (a), SIRI successfully predicts the correct target position, although the copy operation introduces ambiguity; LingUnet, however, predicts the target position in an incorrect opposite direction. Therefore, our proposed SIRI properly characterizes this spatial relationship, whereas LingUnet overfits to the target object localization and causes a significant performance drop .

2) For those samples whose language descriptions have some ambiguous words such as 'next to you', the introduced visual ambiguity misdirects target localization. As shown in Figure 4 (b), both SIRI and LingUnet predict the target position in an incorrect opposite direction, which explains the reason why A@40px for SIRI decreases by around 5%.

## 4.5 Running Time

To evaluate the efficiency of our proposed method, we calculate the number of parameters, running time per 50 images and A@80px for LingUnet and SIRI. It should be noted that theses running times exclude the inference time for feature extraction. All the experiments are conducted with a GeForce GTX TITAN X. As shownn in Table 5, our proposed SIRI cannot be operate at real-time at the moment but some solutions including model compression and model distillation can still be studied. We leave this for future work.

## 5  Conclusion

We present a novel spatial relationship induced network for the SDR task. It characterizes the object-level visual feature correlations, which enables an object to perceive the surrounding scene. Besides, the local spatial relationship is distilled to simplify the spatial relationship representation of the entire image. Further, a global positional information is embedded to alleviate the ambiguities caused by the absence of spatial positions throughout the entire image. Since our proposed network can fully explore these spatial relationships and is robust to the visual ambiguities introduce by a copy-paste operation, our proposed SIRI outperforms the state-of-the-art method LingUnet by 24% for A@80px, and it generalizes consistently well on our proposed extended dataset.

## Acknowledge

This work was supported by the National Key RD Program of China (2018AAA0100704), NSFC(No. 61932020, No. 61773272), the Science and Technology Commission of Shanghai Municipality (Grant No. 20ZR1436000) and ShanghaiTech-Megavii Joint Lab.

## Broader Impact

We would like to claim that our proposed method tackles the spatial description resolution task, under strict privacy preservation by using features rather than raw images. Our work provides novel horizons to the academic community, but it also pushes this task towards real-world application. We also acknowledge that ethical concerns may be caused by the unsatisfactory target localization in some challenging cases, as whis may mislead users. Nonetheless, safety and reliability will be our top priorities when we deploy this system in real-world applications.

## Footnotes

[1]†: Equal Contribution

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
