[Supplementary Material]

# 1 Details about Extended Touchdown Dataset

## 1.1 Extended Touchdown

We build a new extended dataset of the Touchdown, which are collected by the same way as the Touchdown. First, we choose some panorama IDs in the test data of the Touchdown dataset and download the panoramasin equirectangular projection. Then we slice each into eight images and project them to perspective projection. Next we put touchdowns on the target locations in the panoramas and write some language descriptions to instruct people to find them. After that, we also ask some volunteers to double check the annotations by looking for the target with the language we annotate. In addition, these data are collected from the New York StreetView. Although the IDs are the same with ones in the test set of the touchdown dataset, the scene images are changed because of different timestamps. And we rewrite the language descriptions with the new locations of touchdowns, so the dataset is different from the original touchdown dataset. Thus, this new extended dataset will be used to evaluate the generalization as well as visualize the predicted results of our proposed method.

## 1.2 Analysis on the Touchdown and the Extended Touchdown

We further analyze the distribution of orientation words on the Touchdown and the extended Touchdown, as shown in Figure 1. It illustrates a long-tail frequency distribution over the orientation words on both two datasets, where the high-frequency words are quite similar. Also, most of language descriptions contain 20 words on both datasets, which illustrates the consistency of them.

(a) Word Frequency  (b) Length of Language Descriptions

Figure 1: The word frequency and the length of language descriptions on the Touchdown as well as the extended Touchdown.

## 1.3 Examples of Extended Touchdown dataset

*Close to you, there's a fire hydrant and there's a sign in front of the fire hydrant, and this sign is red and yellow, and you click on the center of it to get the location of the touchdown.*

Figure 2: An example of extended touchdown dataset including the panorama and the language description.

# 2 More Results

## 2.1 Results in Successful cases

This part shows the successful examples of SIRI and LingUnet. When both of the methods localize targets correctly, the SIRI is closer to the ground-truth.

*Then stop in front of a parking terminal. Touchdown is on top of the terminal.*

*On your right is an orange newspaper box. Touchdown is on top of the newspaper box.*

*There is a trash can to the right of the tree. Touchdown is sitting on the trashcan.*

*A blue bike locked up on the corner. (the bike should still be in front of you.) You will also see a man with a pink shirt and cane. Touchdown is in the basket of the bike.*

Figure 3: Visualization of predictions of SIRI and LingUnet, as well as the ground-truth. The prediction of SIRI is closer than LingUnet to the ground-truth. Best viewed in color.

## 2.2 Results of Ambiguity only in Images

In this case, there is ambiguity only in images. The language descriptions remove the ambiguity of localization with the global orientations. SIRI can predict correctly because the method can perceive things globally and judge directions like 'your left/right', while LingUnet predicts positions in the opposite locations.

*Touchdown is on top of the mailbox to the left.*

*You should see a cone next to the flowers on the crowd and to the right of the cab. The top of this cone is your touchdown*

*Before you reach the intersection. Look to your left and you will see a white building with grafiiti on it. There is a brown wooden door. To the right of the door on the other side of the railing is a thin window Touchdown is sitting on the ledge of the thin rectangular window to the right of the wooden door.*

*You should see a silver and black hydrant with two posts on either side of it, on your left. Touchdown is on the silver part of the hydrant.*

Figure 4: Results with the ambiguity only in the image where there are same entities. However the language descriptions localized the targets explicitly. SIRI predicts the locations correctly while the LingUnet doesn't.

## 2.3 Results of Ambiguity in both Images and Language descriptions

In this case, the ambiguities in both images and descriptions make it difficult to localize targets correctly. Here are some examples of the failure cases. Although the SIRI and LingUnet predict correctly locally, the final results are wrong because of the ambiguity of the language descriptions.

*Reach the octagonal red and white sign. Near that sign is a red fire hydrant. Touchdown is sitting at the base of this hydrant.*

*Stop next to the post with the small white sign on it on the lefthand side, the chainlink fence is behind it. Touchdown is sitting in the exact middle of the white sign.*

*The yellow building has lamps on both sides of the door. Touchdown is sitting on top of the spike on the lamp to the right of the door.*

*The building is red brick but the door way and window sills and other parts are lighter beige almost yellow. Touchdown is on that top step in the middle going up to the doorway.*

Figure 5: Results with the ambiguities both in the image and language descriptions where there are same entities, and targets can not be localized following the language descriptions. Neither SIRI nor LingUnet predict the location correctly.