[Reviews · NeurIPS 2020]

Review 1

Summary and Contributions: This paper introduces SIRI, an approach for language-guided localization on Streetview panoramic images from the Touchdown dataset. The proposed approach has three key components over a LingUNet base model -- 1) a graph convolutional network based on GloRe units to build richer contextual object representations, 2) a set of convolutional filters explicitly tied to learn spatial relations, one filter per spatial word, activated only if those words are present in the description, and 3) a global coordinate frame for the network to easily pick up on visual-linguistic correlations for words such as "front", "left", "right", etc. Ablation experiments on Touchdown demonstrate significant improvements with the addition of each component, and putting it all together, the final model gets ~20+% absolute improvement across accuracy metrics, setting a new and significantly improved state-of-the-art.

Strengths: This paper presents impressive improvements on a hard problem. This problem is non-trivial because panoramic images have a lot of fine-grained details (that detectors are not typically trained to identify e.g. buildings) and language descriptions in Touchdown typically require multi-hop spatial relational reasoning to localize the target object (unlike visual grounding datasets). The results are impressive because they improve on the previous state-of-the-art by 20+%. The authors also conduct informative ablation experiments to demonstrate the efficacy of each architectural change.

Weaknesses: Conceptually, the local spatial relation guided distillation module is quite similar (but not identical) to FiLM (https://arxiv.org/abs/1709.07871). It would be good to at least discuss this in the text. And including a comparison of the proposed spatial relation guided distillation module vs. an adaptation of FiLM would help situate the architectural contribution better with respect to prior work. Right now it is not clear whether the proposed spatial relation guided distillation module performs that subtask better than other ideas from prior works that could have been used for it (e.g. FiLM).

Correctness: Yes, to my understanding, the empirical evaluations sufficiently demonstrate improved performance on language-guided localization on Touchdown.

Clarity: While the paper is empirically comprehensive, the writing could be significantly smoother and gentler. For example, the intro starts off by listing the challenges in SDR, which is great and helps building intuition for this task, but the following graph talks about object-object correlations, spatial relation words and both of these have not been explained thus far, making it hard to follow. I had to revisit this paragraph after reading the approach section.

Relation to Prior Work: I admittedly am not well-versed with prior works in Touchdown, so not sure if an exhaustive review of prior works has been included. However, with respect to prior works referenced in this paper, the proposed method has novel architectural components for this task and performs significantly better.

Reproducibility: No

Additional Feedback: - Fig 2, "VI" --> "IV"


Review 2

Summary and Contributions: The paper proposes a novel network (spatial relation induced network - SIRI) to tackle to problem of spatial description resolution (SDR), a task that was introduced in Touchdown [2]. For SDR, given an input description and a panoramic image, the goal is to output the (x,y) position on the image that the description refers to. In the prior work, Touchdown, LingUNet was used to fuse the text (encoded using a BiLSTM) and visual features (from a pre-trained ResNet18), and a heatmap of the location is predicted using KL-divergence between the prediction and a Gaussian smoothed distribution of the ground truth location. The main contribution of the paper is the SIRI method, the basic idea of which is to replace the ResNet18 visual features with a combination of three components 1) Use multiple GloRe [3] to encode visual relation features from the image (Object-level Visual Feature Correlation) 2) convolving the GloRe features with filters corresponding to orientation words (the top 6) in the description (Local Spatial Relation Guided Distillation), and 3) viewer specific coordinate embedding with origin at bottom center of the image (Global Spatial Position Embedding) - this looks to be inspired by CoordConv [11]. The authors conduct experiments on the Touchdown SDR task and show that their method outperforms LingUNet. They also conduct ablations showing that each of the components helps to improve the performance. The authors also collect an extended Touchdown dataset.

Strengths: - The specific task, SDR, tackled by the paper is relatively understudied, and experimental results show considerable improvement over the recent SOTA, with each component being useful. - The work would be of interest to researchers at the intersection of vision and language. - As the method is composed of three parts, it's possible that some of those parts can be useful in other vision/language tasks.

Weaknesses: - Part of the method (2), relies on having a list of orientation words. It's not clear how the orientation words are determined. The effect of k (for the number of top-k selected orientation words) is also not studied. - Most of the gains comes from components 2) and 3) and it's not certain how much of usefulness of the method is specific to this dataset. - The ablation study is not as thorough as it can be (it adds the components in order). Ideally, it would also show the effect of 2) and 3) without using 1) (with ResNet features instead of GloRe) and effect of 3) without 2) - The proposed method is very task and data specific and it is not clear whether it would be of interest to the broader NeurIPS community.

Correctness: Mostly correct. I do not see major flaws (beside what is pointed in "Weakness" and "Clarity") The extended Touchdown dataset is collected on the same set of panorama IDs as the original Touchdown test set. As it's essentially the same scenes (albeit at different timestamps and with different instructions), it does weaken the claim of generalization beyond the original Touchdown dataset.

Clarity: The writing is poor at places, making it somewhat challenging to understand certain points and some details hard to follow. Some example segments that were difficult to parse/understand for me: Line 42-43: "each object in the image to perceive the surrounding scene" - the word choice of "perceive" is a bit strange, as the objects don't really perceive Line 101: "the cohesive buildings cannot be differentiated by the detector" Perhaps "the cohesive buildings" => "adjacent buildings" is the desired meaning? Line 109: "features X are over dense, which..." => I'm not sure what is meant by "over dense", perhaps just "dense" is sufficient or perhaps the authors just want to say that both visual features and local spatial relations are included and parts of the sentence "are over dense, which" and "as much as possible" can just be trimmed out? Other Grammar/Spelling/Wording Line 13: "the Touchdown" => "Touchdown" Line 16: "the Touchdown" => "Touchdown" Line 29: "multiply" => "multiple" Line 30: "multiply" => "multiple" Line 167: "This dataset totally" => "This dataset"

Relation to Prior Work: In general, there is limited work on the specific problem of SDR. One work that the authors should discuss (and maybe include in their comparison) is Retouchdown. Retouchdown: Adding Touchdown to StreetLearn as a Shareable Resource for Language Grounding Tasks in Street View, Mehta et al, 2020 There has been extensive work on visual grounding (esp of referring expression) in 2D, where the goal is to output the bounding box or segmentation mask of the object of interest. While these works are briefly discussed, I'm not sure the differentiation is clear and accurate. The key differentiation between SDR and some of the other tasks is that the SDR focuses on identifying a point vs a object. The submission talks about other differences such as "more objects and complex background are included in a large field of view" and that "the provided language descriptions are longer and focused on describing spatial67relation of object pairs". Both are more about the specifics of some of the datasets (such as refcoco+) rather than the specifics of the task.

Reproducibility: No

Additional Feedback: The paper can be improved with 1. Details about how the orientation words are determined (is it from a dictionary?). 2. From section 3.3 (also Figure 2), it was unclear if k or K was the selected number of orientation words. Also in line 120, it was unclear what "H" is. 3. Experiments of the method on the recently released Refer360 dataset Refer360∘: A Referring Expression Recognition Dataset in 360∘ Images, Cirik et al, ACL 2020. See "Weaknesses" and "Clarity" sections for additional suggestions.


Review 3

Summary and Contributions: A new approach to Spatial Description Resolution that achieves new SOTA performance on the Touchdown challenge.

Strengths: The paper presents a novel new approach to interpreting spatial descriptions that achieves an impressive improvement in performance on the Touchdown challenge from Cornell Tech. Experiments are quite comprehensive, including results on a supplementary dataset, ablations, and a manipulated version of the data that introduces additional ambiguities.

Weaknesses: The proposed model is fairly complex and could be better motivated and described. The new extended dataset used in 4.2 should be described more, no details are given on this new data and how it was collected. The broader impacts section is fairly short and could be elaborated and extended.

Correctness: This is a bit hard to judge since it was hard to follow the details of the model, but the method and evaluation appear sound.

Clarity: Paper was fairly hard to follow, partly due to various awkward English expressions. The paper should be proof read and edited by a fluent English speaker. Therefore, I'm not sure I fully understood the complex new model introduced. The paper could better describe the fundamental intuition behind the approach and the motivation for the specific structure of the new proposed architecture.

Relation to Prior Work: Previous work is well covered to my knowledge.

Reproducibility: No

Additional Feedback:


Review 4

Summary and Contributions: This paper proposes a framework called Spatial Relation Induced Network which consists of three parts -- object-level visual correlation, local spatial relation distillation, and global spatial positional embedding. With the proposed three components, the paper achieves 24% improvement over the previous baseline from Touchdown Dataset. Generally, the paper addressed the claim made by the author.

Strengths: The proposed SIRI methods get significant improvement over the LingUnet baseline. In Touchdown and the proposed extended dataset, SIRI exhibits good performance both in qualitative results and localization accuracy. Copy-pasted testing image analysis section is cogent, which demonstrates that SIRI really explores the spatial relation.

Weaknesses: 1) The experiment is somewhat inadequate. In the paper, the author only compares the proposed SIRI approach to the baseline from original Touchdown dataset paper [2]. In fact, spatial description resolution is a similar task as referring expression or instruction grounding. It is necessary for the author to further compare to approaches (such as Mattnet [18] or other new methods in 2019) in those tasks. For example, although Mattnet is not designed for spatial description resolution, but there is also semantic and position modules to handle spatial relation and object relationship reasoning, which can be served as a substitute of Part I & II of SIRI. 2) Although this is a new task, but the solution is compositional and of limited novelty. The proposed framework seems hard to generalize to other tasks besides spatial description resolution since part II and III plays a very important role in SIRI, and the two parts are heavily rely on spatial words. Moreover, the idea to explore spatial relation and reason among objects proposed in the paper is not new and quite prevalent in other vision-language tasks. 3) The ablation study in Table 3 only shows three combinations besides pure LingUnet -- I, I+II and I+II+III, it will be better if the author could provide the combinations of II + III, III only and II only. Then we can better evaluate the importance of Part II and Part III. 4) About the generalization ability of Stage II & III: The global position embedding (Part III) seems to play a quite important role in SIRI. Are this part only limit to Spatial Description Resolution tasks or panoramic images? And if this part can help the navigation task of Touchdown dataset or other referring expressions tasks? For local spatial-word aware gating (Part II), can this part function well in other tasks? 5) The paper has some typos, such as: [Line 92]: adapt should be adopt; [Line 121]: averaged should be summed; [In figure 2]: VI (in fact 6) should be IV (4). If the author(s) could successfully address my above concerns, I will consider to improve my rating after rebuttal. ------------ After rebuttal ------------- After reading the author response and all other reviewers' comments, I think most of my concerns have been well addressed. I would like to improve my overall rating to 6.

Correctness: The method and experiments look correct.

Clarity: The paper is generally readable, but there is large room to improvement on the narrative. Some minor typos also exist in the paper.

Relation to Prior Work: Somewhat discussed the relation to prior works. But still missing discussion of some previous approaches, especially those on instruction grounding, navigation and referring expressions.

Reproducibility: Yes

Additional Feedback:

[Author Response · NeurIPS 2020]

We would like to thank all reviewers for your constructive and intriguing comments. We have concluded more related
work, compared with more SOTA, based on your suggestions. We have significantly polished our paper in terms of
introduction reorganization, grammar correction, fluent expression as well as broader impacts for camera ready.

**Reviewer #1 and Reviewer #4**

Q1: Comparison with FiLM (AAAI 2018). The experiment is somewhat inadequate.

A1: Feature-wise Linear Modulation, namely FiLM is proposed to influence neural network computation via a simple,
feature-wise affine transformation based on conditioning information. Such a novel module has achieved SOTA
performance on the CLEVR benchmark. The GRU module in FiLM functions similar to the BiLSTM module in our
porposed SIRI, and the FiLM functions similar to the spatial relation guided distillation module. Besides, positional
embedding are leveraged in both work. Whereas, the spatial relation guided distillation we introduced induces the
network with a gate mechanism while FiLM performs feature-wise affine transformation. To fairly compare our SIRI
with FiLM, we adopt the part II and the part III only in SIRI. Our SIRI has an accuracy@80px of 58.33%, much better
than FiLM with an accuracy@80px of 52.37%, which demonstrates the effectiveness of our SIRI.

**Reivewer #2**

Q1: The effect of k (for the number of top-k selected orientation words) is also not studied.

A1: The accuracies@80px are 51.02%, 58.33% and 59.74% when k is 4, 6 and 8, respectively.

Q3: The ablation study is not as thorough as it can be (it adds the components in order). Ideally, it would also show the
effect of 2) and 3) without using 1) (with ResNet features instead of GloRe) and effect of 3) without 2).

A3: The accuracy@80px is 56.24% when the component 2) and 3) are adopted without 1). Besides, the accuracy is
38.76% when the component 3) is adopted without 1) and 2).

Q4: It is not clear whether it would be of interest to the broader NeurIPS community.

A4: We believe that our proposed SIRI with strong novelty and promising performance provides a new insight in terms
of architecture design for any vision-language tasks, such as VQA, SDR, etc.

**Reviewer #3**

Q2: The new extended dataset used in 4.2 should be described more, no details are given on this new data and how it
was collected.

A2: Due to the page limitation, the details of the new extended dataset are appended in the supplementray materials.
We also analyze the word frequency on it and visualize the prediction results on this new dataset.

**Reviewer #4**

Q2: Although this is a new task, but the solution is compositional and of limited novelty.

A2: In this paper, we design a novel framework to explicitly tackle the SDR task. Each component is carefully designed
and well investigated. We believe that such a novel framework can push forward this important task and provides a new
insight for any other vision-language task.

Q3: The ablation study in Table 3 only shows three combinations besides pure LingUnet – I, I+II and I+II+III, it will be
better if the author could provide the combinations of II + III, III only and II only. Then we can better evaluate the
importance of Part II and Part III.

A3: The accuracies@80px are 56.24%, 38.76% and 44.95% for II + III, III only, II only, respectively.

Q4: About the generalization ability of Stage II & III.

A4: Because MAttNet detects objects and scores the RoIs, the Stage II and III in our paper are not suitable to MAttNet.
We use YOLO-VG(A Fast and Accurate One-Stage Approach to Visual Grounding, ICCV2019) as our baseline, a one
stage method for visual grounding, which can be end-to-end trained. To investigate the generalization ability of Stage II
& III, we add the Stage II and III to it. The results show that Stage II and III can improve the performance by 0.9% and
0.7%, respectively. Thus, our proposed modules can also perform well on other tasks and datasets.

Q5: The paper has some typos, such as: [Line 92]: adapt should be adopt; [Line 121]: averaged should be summed; [In
figure 2]: VI (in fact 6) should be IV (4).

A5: Thanks for pointing out these typos. We have significantly polished this paper for camera ready.

[Meta-Review · NeurIPS 2020]

Paper was reviewed by four expert reviewers, with initial scores of: 6, 6, 6, 5. Reviewers acknowledge a commendable improvements of the proposed approach on a difficult and novel task. A number of issues where raised about the paper, including (1) poor exposition and language [all reviewers], (2) lack of comparison to FiLM [R1], (3) specificity of task and dataset [R2,R4], among others. Authors provided a rebuttal that was discussed by reviewers and ultimately convincing. Two of the reviewers upgraded their scores, resulting in unanimously positive, albeit marginally so, scores of: 7, 6, 6, 6. AC, despite having reservations about quality of the writing, mentioned by all reviewers, agrees that the approach is valuable and presents a significant improvement over state-of-the-art on a relatively unexplored problem. As such AC agrees with reviewers that the paper should be accepted. Authors are asked to make a significant attempt to improve the writing and the language for the camera ready.